# Improving the Reliability of Large Language Models by Leveraging Uncertainty-Aware In-Context Learning

## Abstract

In recent years, large-scale language models (LLMs) have gained attention for their impressive text generation capabilities. However, these models often face the challenge of "hallucination," which undermines their reliability. In this study, we introduce an uncertainty-aware in-context learning framework to empower the model to enhance or reject its output in response to uncertainty. Human-defined methods for estimating uncertainty typically assume that "uncertainty is lower when the model's response is correct compared to when it is incorrect." However, setting a precise threshold to distinguish correctness is challenging. Therefore, we introduce uncertainty information as an intermediary variable that implicitly influences the model's behavior. Our innovative uncertainty-aware in-context learning framework involves fine-tuning the LLM using a calibration dataset. Our aim is to improve the model's responses by filtering out answers with high uncertainty while considering the model's knowledge limitations. We evaluate the model's knowledge by examining multiple responses to the same question for the presence of a correct answer. When the model lacks relevant knowledge, the response should indicate that the question cannot be answered. Conversely, when the model has relevant knowledge, the response should provide the correct answer. Extensive experiments confirm the effectiveness of our framework, leading to two key findings. First, the logit output values of the LLM partly reflect inherent uncertainty. Second, our model autonomously recognizes uncertainty, resulting in improved responses.

## 1 Introduction

Despite the remarkable progress made in the field of Large-scale Language Models (LLMs) (He & Garner (2023) Deng & Lin (2022) Aljanabi et al. (2023)), they remain susceptible to well-known reliability issues, particularly in the domain of trustworthiness regarding their generated content. In the context of language models, "hallucination" refers to the generation of text or responses that appear to be syntactically sound, fluent, and natural but are factually incorrect, nonsensical, or deviate from the provided source input (McKenna et al. (2023)).

To alleviate the hallucination problem, an emerging research direction is uncertainty estimation for LLMs. They seeks to model the inherent uncertainty in LLM responses, primarily by analyzing the logit output values of generated tokens (Huang et al. (2023) Xiong et al. (2023) Kadavath et al. (2022)). The fundamental hypothesis driving uncertainty estimation is that "uncertainty is lower when the model's response is correct compared to when it is incorrect." While existing works aim to develop more effective uncertainty estimation criteria to align with this hypothesis, the question of how to optimize model responses based on the calculated uncertainty remains relatively unexplored within the research community.

In this work, we focus on the problem of refining the LLM responses based on the calculated uncertainty. Setting a strict uncertainty threshold to discern the correctness of the model's response seems intuitive, but it presents a formidable challenge in practice. Hence, we turn to propose a novel uncertainty-aware in-context learning framework. We encode uncertainty information as an intermediary variable that can implicitly influence the model's behavior. Concretely, our goal centers on

the adaptive adjustment of the model's responses in cases of high uncertainty. We believe that when the model generates a response with high uncertainty and it is subsequently assessed as incorrect, the response should be modified to the correct answer if the model possesses the necessary knowledge. Conversely, if the model lacks the necessary knowledge, the response should be altered to indicate that it cannot answer the question. Besides, for responses with low uncertainty but deemed incorrect, which we refer to as "over-confidence problem," we believe that this indicates a deficit in the model's knowledge. It is worth noting that the over-confidence problem is not our target scenario and cannot be corrected through uncertainty-based modifications.

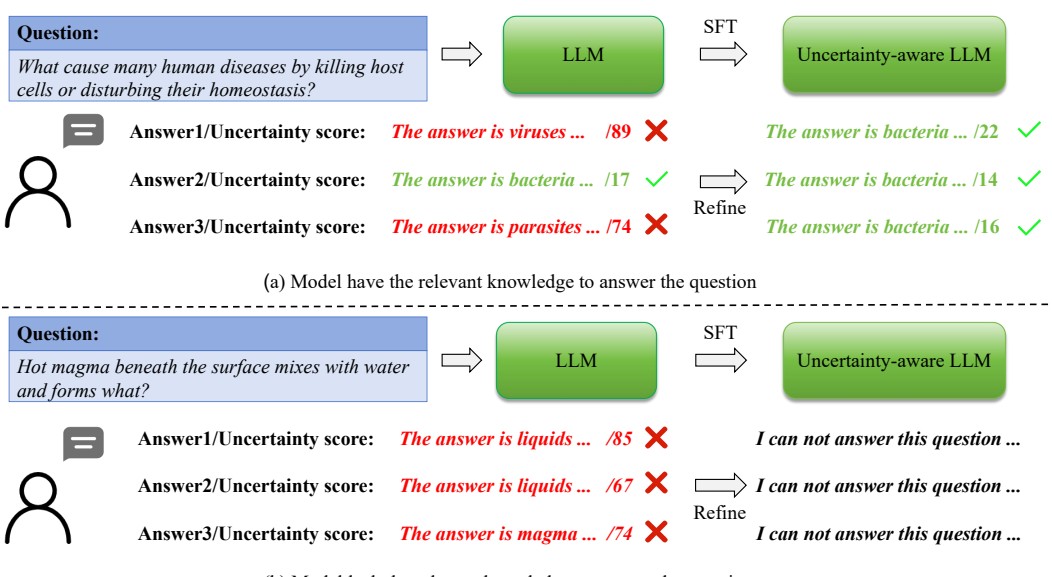

Figure 1: An illustration of automatic correction of LLM responses based on uncertainty score. Two scenarios are considered: firstly, the model has the relevant knowledge (illustrated in Figure (a)), and secondly, the model lacks relevant knowledge (illustrated in Figure (b)). If the model has the relevant knowledge, the incorrect response should be modified to the correct answer . Conversely, if the model lacks the relevant knowledge, the response should be altered to indicate that it cannot answer the question.

To achieve the above goals, we propose an uncertainty-aware in-context learning framework. Our framework incorporates the use of a calibration dataset to fine-tune the LLM. This fine-tuning procedure is designed to preserve the model's linguistic fluency while equipping it with the ability to adapt its responses in response to uncertainty. The calibration dataset is comprised of question-answer pairs, where each question is equipped with one correct answer along with multiple distractor options. In our framework, for each question in the calibration dataset, the model generates multiple responses, each labeled as "correct" or "incorrect", with corresponding uncertainty calculations. When all of the model's responses for a particular question are classified as "incorrect", the uncertainty-aware model should refrain from providing an answer. Conversely, if at least one of the model's responses aligns with the correct answer, the uncertainty-aware model should select the correct response as its final answer. This process ensures that the model dynamically adjusts its behavior in response to the inherent uncertainty, enhancing its overall reliability.

To summarize, we make a comprehensive analysis of existing uncertainty estimation methods, which we seamlessly integrate into our uncertainty-aware in-context learning framework. We introduce a promising strategy for leveraging uncertainty to enhance the reliability of LLM responses. And it is demonstrated through extensive experiments that our framework delivers obviously performance improvement on the test set. We draw two main conclusions from our experimental results. First, the token generation probability of LLM partially reflects the inherent uncertainty. Second, our framework has the ability to recognise uncertainty autonomously, thus improving its response. We also demonstrate that the model's behavior can be effectively modified by introducing uncertainty

information during inference stage. This dynamic adaptation empowers the model to flexibly refine its responses in alignment with its knowledge boundaries.

# 2 METHOD

## 2.1 OVERVIEW

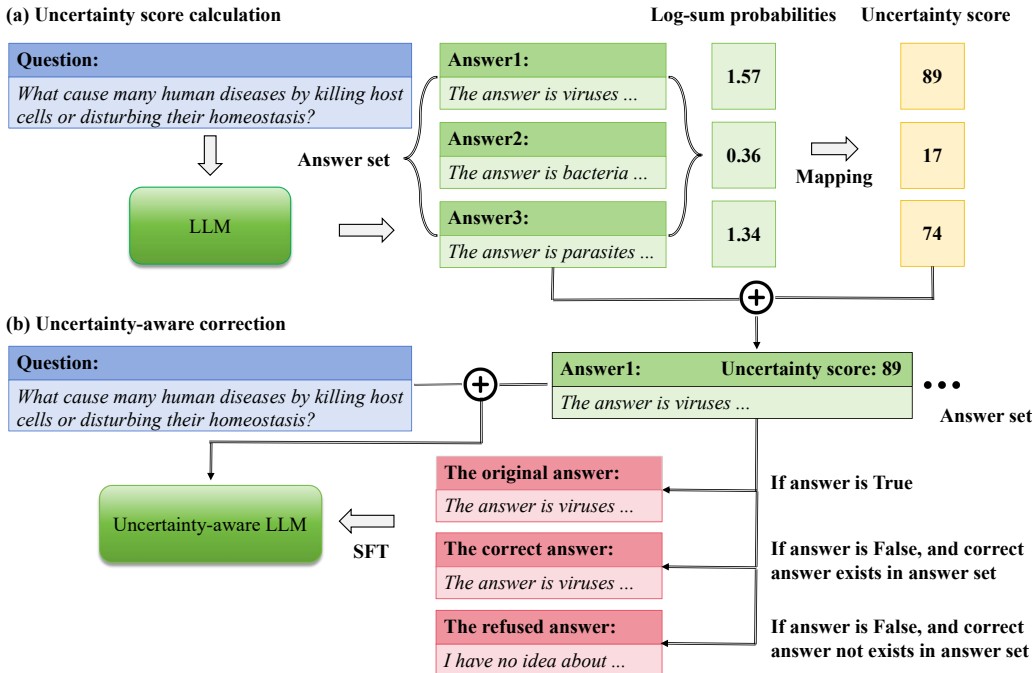

Figure 2: An illustration of our uncertainty-aware framework. It reveals a two-step process comprising uncertainty score calculation and uncertainty-aware correction. In the first step, known as uncertainty score calculation, we undertake two subtasks: uncertainty estimation and uncertainty score mapping. This sequence results in the generation of an uncertainty score for each response within the answer set. Subsequently, we incorporate these uncertainty scores into the LLM, facilitating adaptive self-correction. Through a supervised fine-tuning procedure, we empower the model to explore and expand its knowledge boundaries. This fine-tuning process enables the model to autonomously refine its responses by taking into account the calculated uncertainty scores.

Figure 3 shows an overview of our framework. Our uncertainty-aware framework is divided into two steps, namely uncertainty score calculation and uncertainty-aware correction. To compute the uncertainty score, we follow existing works in uncertainty estimation of LLM. Firstly, we utilize the logit output values of the model's response to obtain the uncertainty of each generated token. Then we aggregate these token-level uncertainties to derive the uncertainty of each generated output. We will elaborated this part in Section 2.2.1.

To explore the knowledge boundary of the model and investigate the beam search space within the answer set, we make the model responds to the same question multiple times. For each of these responses, we calculate the uncertainty as mentioned. Then, we map the uncertainty of each response to an uncertainty score, drawing upon the model's uncertainty distribution on the training set. We will elaborated this part in Section 2.2.2.

After obtaining the uncertainty score for each response, we supervised fine-tuning the model for self-correction. This fine-tuning process empowers the model to autonomously adjust its responses based on the calculated uncertainty scores. The format of the SFT dataset and finetuning procedure will be elaborated in Section 2.3.1.

Finally, we introduce the test-time correction with our uncertainty-aware framework in Section 2.3.2.

## 2.2 Uncertainty Score Calculation

### 2.2.1 Uncertainty estimation method

Given an input $\mathbf{X} = [x_1, x_2, \ldots, x_n]$ ($x_i$ denotes $i$-th input token), an LLM $f$ with pre-trained weights $w$. The output $\mathbf{Y} = [y_1, y_2, \ldots, y_m]$ ($y_j$ denotes $j$-th generated token) is generated through a decoding process. An uncertainty estimation method $g$ is to calculate a score $s$ regarding the uncertainty of $Y$. In general, uncertainty estimation methods are classified into single-inference based and multi-inference based. To improve the efficiency of the inference stage, our framework focuses on the former. We will therefore focus on the former and briefly introduce the latter.

**Single-inference Based Uncertainty Estimation.** Single-inference based uncertainty estimation methods model the token-level logit output value to obtain the sentence-level uncertainty. First, we apply the softmax function over the logit output values for each token $y_j$ and obtain maximum softmax probabilities $[p_1, p_2, \ldots, p_m]$. We study the four standard techniques for calculating uncertainty based on the probabilities.

**Minimum of Token Probabilities.**

$$s = -\log(Min(p_1, p_2, \ldots, p_m)) \tag{1}$$

In this way, we use the minimum token probabilities to represent the whole sentence.

**Average of Token Probabilities.**

$$s = -\log(Avg(p_1, p_2, \ldots, p_m)) \tag{2}$$

In this way, we directly average the probabilities before feeding it into the logarithmic function.

**Normalized Product of Token Probabilities.**

$$s = -\log\left(p_1 \cdot p_2 \cdot \ldots \cdot p_m\right)^{\frac{1}{m}} \tag{3}$$

In this way, we take a normalized product of the probabilities of each token.

**Log-sum of Token Probabilities.**

$$s = -\log\left(p_1 \cdot p_2 \cdot \ldots \cdot p_m\right) \tag{4}$$

In this way, we take the Log-sum of each token probabilities to represent uncertainty. We will give experimental results and analysis of these four techniques in the Experiment Section 3.1.

**Multi-inference Based Uncertainty Estimation.** Multi-inference based uncertainty estimation methods first collect a answer set through test-time data augmentation methods. Then the uncertainty is estimated as the divergence among those predictions. Concretely, if the more semantically similar a response is to other responses, the lower the uncertainty of that response. Because our framework involves utilize the uncertainty for self-correction, whereas multi-inference based uncertainty estimation methods need to get the answer set before calculating the uncertainty. This will significantly increase the overhead of the inference stage. Hence, we do not use these uncertainty estimation methods in our framework.

### 2.2.2 Uncertainty score Mapping

To enable the model to generate adaptive corrective responses based on uncertainty, we reintroduce the computed uncertainty into the model by incorporating it as contextual information. Due to the highly uneven distribution of uncertainty in the training dataset, the direct inclusion of uncertainty introduces a great deal of noise and confusion.

To mitigate the challenges posed by the non-uniform distribution of uncertainty within our dataset, we adopt a strategy of mapping uncertainty values to a discrete set of labels, referred to as "uncertainty scores." In our experiment, we investigate the impact of various classification granularities, namely deciles, hundreds, and thousands, which correspond to uncertainty scores ranging from 1 to 10, 1 to 100, and 1 to 1000, respectively.

Our approach involves uniformly partitioning the uncertainty distribution within the training dataset into a matching number of intervals based on the chosen granularity for classification. Within each

interval, uncertainties are assigned identical uncertainty scores. This same mapping strategy is applied during the inference stage, with uncertainty scores computed based on the intervals established during the training phase.

This mapping technique serves to alleviate the complexity associated with interpreting uncertainty within the model, leading to improvements in both robustness and overall performance.

### 2.3 UNCERTAINTY-AWARE CORRECTION

#### 2.3.1 UNCERTAINTY-AWARE FINETUNING

After obtaining the uncertainty scores for each response, we utilize the calibration dataset to construct an uncertainty labelled fine-tuned dataset to train the model. Specifically, for each question in the calibration dataset, it was paired with one correct option and multiple distractor options. It's essential to emphasize that our framework is primarily centered around uncertainty-aware correction rather than the infusion of fresh knowledge into the model. Consequently, rather than directly injecting correct answers from the calibration dataset into the model, we employ it to assess the correctness of the model's responses.

In our framework, for each question in the calibration dataset, the model first generates multiple responses. Then we categorize each response as either "correct" or "incorrect" depends on the correct answer and calculating corresponding uncertainty scores as mentioned. In cases where all of the model's responses for a particular question are categorized as "incorrect," we interpret this as an indication that the model lacks the requisite knowledge to respond to the question. Consequently, we assign the ground truth for such instances to a rejection response, such as "I am unable to answer the question due to a lack of relevant knowledge." On the other hand, when at least one of the model's responses aligns with the correct answer, we infer that the model possesses the requisite knowledge to address the question. In such scenarios, for the responses classified as incorrect under this question, we designate the ground truth as a randomly selected correct answer from the answer set of this question. Conversely, for the response classified as correct under this question, the ground truth remains unchanged, *i.e.*, it retains the original response. After constructing the fine-tuned dataset, we follow a standard supervised fine-tuning process to train the uncertainty-aware LLM.

#### 2.3.2 TEST-TIME CORRECTION

During inference stage, to improve the efficiency of reasoning, we no longer answer each question multiple times to get a answer set. Instead, the model respond to each question once and calculate its corresponding uncertainty score. We then resend the problem, the response and the uncertainty score to the model for self-correction.

Despite the fact that our framework also introduces a increased inference cost, we contend that, given the current state of technology, it is of greater significance to prioritize the resolution of reliability concerns of Language Models (LLMs).

## 3 EXPERIMENTS

### 3.1 EXPERIMENTAL SETTINGS

**Datasets.** We consider the free-form question answering datasets SciQ (Welbl et al. (2017)) as the calibration dataset, which is widely used in the uncertainty estimation task. In our experiments, we utilize the validation set (1,000 questions) of SciQ as the testset and the training set of SciQ to construct training set.

**Evaluation Metrics.** In our framework, the model categorizing questions into two distinct groups: "refused questions" and "answered questions." Within this context, we employ the term "accuracy" (abbreviated as "acc") to quantify the percentage of correct responses specifically among the "answered questions." Additionally, we introduce the concept of the "answer rate," which serves as a measure denoting the proportion of "answered questions" in relation to the total number of questions. To determine whether a response to a question is correct, we first input a prompt "Choose

a correct answer," and we subsequently use the ChatGPT to extract the answer from the model's response and check its correctness.

To evaluate the merits and drawbacks of various uncertainty estimation methods concerning their alignment with the fundamental hypothesis that "uncertainty is lower when the model's response is correct than when it is incorrect," we employed the AUROC metric. The AUROC metric is known as the "Area Under the Receiver Operator Characteristic Curve" (AUROC). It is equivalent to the probability that, when selecting answers at random, a correct response will possess a higher uncertainty score than an incorrect response.

**Implementation Details.** To construct answer set, we generate 5 responses for each question. The temperature t is set to 0.001. We conduct experiments on the Vicuna (Chiang et al. (2023)) and LLama (Touvron et al. (2023)) model. We repeat each experiment five times and report the mean of results. All the experiments are conducted with 8 NVIDIA A100 GPUs.

## 3.2 EXPERIMENTAL RESULTS

Our experimental results is provided in Table 1.

**Analysis of different uncertainty estimation methods.** We first give a analysis for different uncertainty estimation methods. The "Min" approach, in practice, selects the token with the highest uncertainty within a response to represent the overall uncertainty of the entire response. However, it is noteworthy that responses frequently consist of multiple tokens characterized by high uncertainty values, potentially leading to information loss.

On the other hand, methods denoted as 'Avg" and "Norm" adopt a different strategy for estimating the uncertainty of the complete response. They calculate a form of mean uncertainty by considering each token within the response. However, it is important to acknowledge that not every token in a response carries equal significance in terms of uncertainty. In some cases, the uncertainty associated with certain tokens, particularly those deemed as meaningless words or prepositions, may not be of paramount importance. During the averaging process, it is essential to recognize that the presence of numerous tokens categorized as such—those with low informational content—can dilute the tokens with high uncertainty.

The "Log-sum" method, while viable, presents its own set of limitations. As it combines the uncertainty values of all tokens within a response to represent the overall response uncertainty, it can be notably influenced by the response's length. To mitigate this length-based bias, we adopt a practice of computing uncertainty solely for the sentence in the response that holds a pivotal role in the decision-making process. Typically, this is the initial sentence, aligned with our prompt setting.

To ensure fair comparison among different uncertainty calculation methods, we maintain consistency by calculating uncertainty exclusively for the sentence that drives the decision-making process for other methods as well.

It can be observed from the Table 1 that the "Log-sum" uncertainty estimation methods achieves best performance in our framework. When incorporating with LLama, our framework delivers a 10.3% performance improvement compared to "LLama-finetuned" in terms of accuracy. When incorporating with Vicuna, our framework delivers a 9.0% performance improvement compared to "Vicuna-finetuned" in terms of the Accuracy. This demonstrates that the reliability of the model responses is obviously improved by injecting uncertainty information.

**Analysis of mean uncertainty score and AUROC.** As depicted in the Table 1, it is evident that our framework's exhibits a notable reduction in mean uncertainty score, accompanied by an improvement in accuracy. This observation underscores the presence of a correlation between the mean uncertainty of responses and overall accuracy. This further supports the hypothesis that "uncertainty is lower when the model's response is correct than when it is incorrect."

Furthermore, our analysis reveals a positive association between the AUROC metric and the accuracy of the method. This relationship suggests that a superior uncertainty estimation method can enhance the validity of our framework.

Table 1: Experimental results on SciQ dataset. "LLama" and "Vicuna" denotes the LLM backbone of our framework. "Min","Avg","Norm","Log-sum" denotes the four different uncertainty estimation methods "Minimum of Token Probabilities","Average of Token Probabilities","Normalized Product of Token Probabilities","Log-sum of Token Probabilities", respectively. "finetuned" denotes directly finetuning the backbone using the same training set. The best accuracy performance is shown in bold.

| Method | Accuracy | Answer rate | Accuracy*Answer rate | Mean uncertainty score | AUROC |
|---|---|---|---|---|---|
| LLama (Touvron et al. (2023)) | 44.5 | 100.0 | 44.5 | 49.3 | – |
| LLama-finetuned | 66.9 | 100.0 | 66.9 | 45.2 | – |
| Ours-LLama(Min) | 75.0 | 89.3 | 67.0 | 27.8 | 68.1 |
| Ours-LLama(Avg) | 73.8 | 90.5 | 66.8 | 28.0 | 67.3 |
| Ours-LLama(Norm) | 74.1 | 89.4 | 66.2 | 28.4 | 67.9 |
| Ours-LLama(Log-sum) | **77.2** | 88.1 | **68.0** | 27.6 | 70.2 |
| Vicuna (Chiang et al. (2023)) | 67.1 | 100.0 | 67.1 | 46.1 | – |
| Vicuna-finetuned | 84.6 | 100.0 | 84.6 | 44.0 | – |
| Ours-Vicuna(Min) | 92.2 | 94.5 | 87.1 | 22.3 | 70.0 |
| Ours-Vicuna(Avg) | 91.3 | 95.0 | 86.7 | 24.6 | 69.5 |
| Ours-Vicuna(Norm) | 91.7 | 95.2 | 87.3 | 23.9 | 68.8 |
| Ours-Vicuna(Log-sum) | **93.6** | 94.4 | **88.4** | 21.4 | 70.5 |

Table 2: Examples of our uncertainty-aware correction of model Vicuna on the testset. The color green denotes correct responses, while red denotes incorrect ones. The "Original" denotes original responses while "Refine" denotes uncertainty-aware correction responses. For ease of expression, we only display the initial sentence from the complete response.

| Question/type | Answer set |
|---|---|
| The olfactory receptor neurons are located in a small region within what cavity? / Original | 1.The olfactory receptor neurons are located in the olfactory epithelium. Uncertainty score: 96
2.The correct answer is between inferior nasal and posterior nasal. Uncertainty score: 45
3.The correct answer is: the superior nasal cavity. Uncertainty score: 51 |
| The olfactory receptor neurons are located in a small region within what cavity? / Refine. | 1.The olfactory receptor neurons are located in the superior nasal cavity. Uncertainty score: 16
2.The correct answer is: superior nasal, Uncertainty score: 9
3.The correct answer is: the superior nasal cavity. Uncertainty score: 11 |
| What type of response is generated when a stimulus is received by the nervous system?/ Original | 1.The correct answer is an action potential. Uncertainty score: 85
2.The correct answer is: A neural response. Uncertainty score: 91
3.The correct answer is: a motor response. Uncertainty score: 63 |
| What type of response is generated when a stimulus is received by the nervous system?/ Refine | 1.Based on the above information, I have no idea on this question.
2.The correct answer is: a motor response. Uncertainty score: 21
3.The correct answer is: a motor response. Uncertainty score: 15 |
| Magnetic poles always occur in pairs - what are the names of each pole? / Original | 1.The correct answer is "southwest and southeast". Uncertainty score: 65
2.The correct answer is "tropic and arctic". Uncertainty score: 82
3.The correct answer is "southwest and southeast". Uncertainty score: 59 |
| Magnetic poles always occur in pairs - what are the names of each pole? /Refine | 1.Based on the above information, I have no idea on this question.
2.Based on the above information, I have no idea on this question.
3.Based on the above information, I have no idea on this question. |

## 3.3 CASE STUDIES

As shown in Table 2, our model employs adaptive self-correction strategies based on the uncertainty score. From the case presented in the table, there are three main observations:

First, it is evident that the uncertainty scores associated with correct responses do not consistently fall below those of incorrect responses. Nonetheless, our framework exhibits robustness in its ability to retain correct responses and rectify incorrect ones.

Second, during the training phase, the model only rejects responses when there is no correct answer available within the answer set. In contrast, during testing, responses with high uncertainty scores may be rejected, even if the model has the necessary knowledge to generate an answer. This adjustment does not necessarily improve the accuracy of the model's responses, but it does improve the reliability of the model's responses.

Third, the model consistently modifies its behavior to reject responses when it encounters questions that fall beyond its knowledge domain.

Figure 3 illustrates the transformation in model behavior as a result of uncertainty-aware fine-tuning. Our analysis involved tracking changes in decision outcomes across four categories: "True2False," "True2Unknown," "False2True," and "False2Unknown." Notably, the figure reveals a obviously discrepancy, with a significantly lower count of decisions exhibiting negative shifts compared to those showing positive shifts.

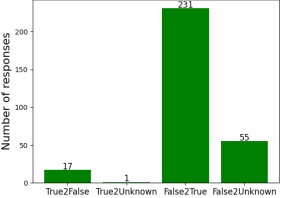
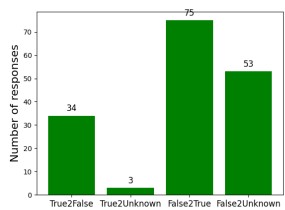

(a) Changes in the behavior of our model in contrast to Vicuna

(b) Changes in the behavior of our model in contrast to Vicuna-finetuned

Figure 3: An illustration of the change in model behavior as a result of uncertainty-aware fine-tuning. "Our-model" refer to the "Ours-Vicuna(Log-sum)" mentioned before.

This observation demonstrates the consistent benefits of introducing uncertainty information into the model. It also supports the potential for enhancing the model's reliability through our framework.

## 4 DISCUSSIONS

**Q.1 Is there any better uncertainty estimation methods?** Uncertainty estimation methods play a pivotal role in our framework, and our experimental findings unequivocally demonstrate that the quality of uncertainty estimation directly correlates with overall performance. A natural concept that arises is to initially extract entities from responses and subsequently compute uncertainty values for each entity independently. However, this approach is not without its challenges, as the extraction of entities itself can introduce errors, and not all responses inherently focus on entities.

Furthermore, our investigation has revealed an important distinction between the manner in which current uncertainty computation methods operate and how human comprehension functions. While existing methods compute sentence-level uncertainty based on token-level generation probabilities, human comprehension operates primarily at the word level, not the token level. In fact, it is often the initial token that significantly influences the overall comprehension. Therefore, a more rational approach appears to be modeling token-level generation probabilities to derive word-level confidence. Subsequently, this word-level confidence can be used to model the uncertainty of the entire sentence, predicated on the semantic understanding of the words within that sentence.

**Q.2 How to deal with over-confidence problem?** In addressing the issue we referred to as "over-confidence problem," characterized by low uncertainty in model responses that are nonetheless incorrect, our analysis identifies two principal contributing factors. Firstly, we argue that the uncertainty within this study pertains to the model's confidence level regarding its responses. In situations where the model has gaps in knowledge or expertise, it can exhibit a high degree of unwarranted confidence in providing incorrect answers. Furthermore, while the probability of the generated token can serve as a partial indicator of the model's uncertainty, it does not encapsulate the full spectrum of uncertainty. Our observations reveal that certain over-confidence questions are highly susceptible to variations in question content, resulting in significant fluctuations in model output. Consequently, we believe that the model's resilience to variations in question content may serve as a potential indicator to judge whether a given response is a over-confidence question or not.

**Q.3 Data uncertainty and model uncertainty.** The scope of our study centers on the model's confidence level regarding its responses, specifically focusing on the concept of model uncertainty. It is crucial to acknowledge that inherent uncertainties exist within the original data or the questions posed. Within this context, certain questions are open-ended, yielding non-unique answers, while others inherently lack a definitive response, as exemplified by inquiries like "Are there aliens?" We classify this category of uncertainty as data uncertainty.

In scenarios where data uncertainty prevails, the model's response confidence does not necessarily correlate directly with the uncertainty surrounding the answer. As such, within the framework of our research, we intentionally target for datasets and scenarios where there exists a single, unequivocal

answer. This enables us to maintain precision and clarity in assessing the model's confidence without confounding factors arising from the intrinsic uncertainty present in the data or questions themselves.

## 5 RELATED WORKS

### 5.1 UNCERTAINTY ESTIMATION

Uncertainty estimation has been a subject of extensive study over the years. Glushkova et al. (2021) employed techniques such as Monte Carlo dropout to observe the effect of model perturbations on output. Fomicheva et al. (2020) utilized uncertainty quantification methods to improve probability estimates in neural networks for better quality estimation. Ott et al. (2018) assessed uncertainty by comparing multiple model outputs against multiple references, using inter-sentence BLEU as a metric. Lahlou et al. (2021) introduced a model-agnostic framework called Direct Epistemic Uncertainty Prediction for estimating epistemic uncertainty in machine learning models. In the context of regression tasks, Wang et al. (2022) employed uncertainty estimation to address both data and model uncertainty, while Malinin et al. (2020) proposed a method for uncertainty estimation using Prior Networks, which provides interpretable measures of uncertainty at a low computational cost. For Natural Language Understanding tasks, Talman et al. (2023) applied Bayesian uncertainty modeling using Stochastic Weight Averaging-Gaussian.

Besides, Large-scale Language Models (LLMs), including ChatGPT, are known to be hallucinatory, generating counterfactual responses, which may affect the credibility of their outputs (Manakul et al. (2023)). In the realm of uncertainty estimation within Large Language Models (LLMs), it's noteworthy to acknowledge that while extensive research has delved into models with distinct labels, such as classification models (Ulmer et al. (2022)), the domain of uncertainty estimation remains relatively under-explored in the context of popular free-form LLMs like LLama (Touvron et al. (2023)) and Vicuna (Chiang et al. (2023)). Uncertainty estimation in these models introduces a unique challenge, primarily due to their adaptable and virtually limitless solution spaces. In this context, any generated output can conceivably be deemed correct, provided it consistently aligns with the underlying semantics of the actual answer.

Recent advancements in this field have witnessed notable contributions. For instance, Xiao et al. (2022) conducted comprehensive empirical evaluations, exploring the impact of various configurations, including model size, architecture, and training loss, on uncertainty within LLMs. Meanwhile, Lin et al. (2022) proposed novel methodologies aimed at quantifying uncertainty by directly prompting language models to express uncertainty about their generated responses.

More recently, Predictive Entropy (Kadavath et al. (2022)) and Length-normalized Predictive Entropy (Malinin & Gales (2020)) calculates the uncertainty of its response by using the probability of generated each token. Semantic Entropy (SE) (Kuhn et al. (2023)) has emerged as a promising approach to tackle the challenge of "semantic equivalence" in uncertainty quantification. SE operates by clustering generated responses that share the same semantics and employs cluster-wise predictive entropy as a means of measuring uncertainty.

In contrast to the aforementioned methods, in this work, our primary objective is not the development of improved uncertainty estimation techniques for Large Language Models (LLMs). Instead, our focus is on leveraging uncertainty as a potential information to dynamically refine and enhance the model's generated responses.

## 6 CONCLUSIONS

To mitigate the issue of hallucination in Lagre Language Models (LLM), we present an innovative framework known as Uncertainty-Aware In-context Learning. Our framework involves fine-tuning the LLM using a calibration dataset, with the objective of preserving linguistic fluency while endowing it with the capability to adapt responses based on uncertainty. Our extensive experimentation substantiates the efficacy of our framework and leads to two noteworthy findings. First, we observe that the logarithmic output values of the LLM partially reflect inherent uncertainty. Second, our model exhibits autonomous recognition of uncertainty, resulting in improved response accuracy.

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
