# OpenReview forum: "Improving the Reliability of Large Language Models by Leveraging Uncertainty-Aware In-Context Learning"
_ICLR.cc/2024/Conference — Submitted to ICLR 2024_

### Official Review · Reviewer_9t2n · 2023-10-31

**Soundness:** 2 fair
**Presentation:** 2 fair
**Contribution:** 2 fair
**Rating:** 3
**Confidence:** 4

**Summary:**

This paper introduces an innovative uncertainty-aware in-context learning framework to enable LLMs to enhance or reject their output in response to uncertainty. Unlike human-defined methods that struggle with setting precise correctness thresholds, this framework introduces uncertainty information as an intermediary variable to influence the model's behavior. The approach involves fine-tuning the LLM using a calibration dataset and aims to improve responses by filtering out answers with high uncertainty while considering the model's knowledge limitations. The evaluation confirms that the logit output values of the LLM partially reflect inherent uncertainty, and the model autonomously recognizes uncertainty, leading to improved responses.

**Strengths:**

1. The studied problem of reliability of LLMs is important and holds significant importance in applying LLM to real-life scenarios.

2. The authors validate the effectiveness of the uncertainty-aware in-context learning framework.

**Weaknesses:**

1. The experiments are only conducted on one dataset, limiting the generalization of the proposed approach.

2. It is unclear how the 'calibration dataset' is created in section 2.3.1. The way to determine the range of uncertainty (ranging from 1 to 10, 1 to 100, and 1 to 1000) is arbitrary without sufficient empirical or theoretical support.

3. Some experiment details are not enough. The authors only compare the model with vanilla fine-tuning. Can some of the methods mentioned in the paper [1] be used as baselines?

4. There are some typos in the paper. For example, in section 2.1, 'We will elaborated this part' should be 'We will elaborate this part'.

5. Missing references. Many papers discussing on LLM uncertainty are missing such as [2,3,4].

References:

[1] Xiong, Miao, et al. "Can LLMs Express Their Uncertainty? An Empirical Evaluation of Confidence Elicitation in LLMs." arXiv preprint arXiv:2306.13063 (2023).

[2] Lin, Stephanie, Jacob Hilton, and Owain Evans. "Teaching models to express their uncertainty in words." arXiv preprint arXiv:2205.14334 (2022).

[3] Kuhn, Lorenz, Yarin Gal, and Sebastian Farquhar. "Semantic uncertainty: Linguistic invariances for uncertainty estimation in natural language generation." arXiv preprint arXiv:2302.09664 (2023).

[4] Quach, Victor, Adam Fisch, Tal Schuster, Adam Yala, Jae Ho Sohn, Tommi S. Jaakkola, and Regina Barzilay. "Conformal Language Modeling." arXiv preprint arXiv:2306.10193 (2023).

**Questions:**

1. In section 3.1, why do you set the temperature to 0.001?

2. Could you provide additional insights into the reasons behind the procedure for constructing the calibration dataset?

3. Would it be possible to expand the scope of datasets used in your experiments?

---

### Official Review · Reviewer_JPa6 · 2023-11-01

**Soundness:** 3 good
**Presentation:** 2 fair
**Contribution:** 3 good
**Rating:** 3
**Confidence:** 3

**Summary:**

The paper describes an uncertainty aware in-context learning (ICL) framework to help fine-tune LLMs. The ICL framework together with different uncertainty estimation methods are used to assess correctness of the LLM's responses and empirically show improvements in overall performance.

**Strengths:**

The idea presented is simple and interesting. Incorporating uncertainty does show significant improvements when compared to the base LLM model or its fine-tuned variant.

**Weaknesses:**

1) The paper is difficult to read and understand. It would benefit from a few more rounds of writing revisions.

2) The paper lacks methods to compare against. While the proposed method does better than the base model and its fine-tuned version, it would good to include how close or better the proposed approach is with respect to SOTA methods on the target dataset.

3) The authors do not discuss about granularity of the uncertainty scores and its impact on performance which they said they did in the experiments section on page 4.

4) Please correct typos and grammatical mistakes. Certain abbreviations are referenced before they are defined and some aren't defined at all, e.g., SFT in Figure 1 on page 2.

**Questions:**

Please see above comments.

---

### Official Review · Reviewer_4TCY · 2023-11-03

**Soundness:** 1 poor
**Presentation:** 2 fair
**Contribution:** 1 poor
**Rating:** 3
**Confidence:** 3

**Summary:**

This paper proposes a framework for improving the reliability of large language models (LLMs) by leveraging uncertainty-aware in-context learning. It proposes a supervised fine-tuning procedure that enables the LLM to adaptively adjust its responses based on the uncertainty scores, either by refining incorrect answers or by refusing to answer when the model lacks relevant knowledge. Extensive experiments on the SciQ dataset demonstrate the effectiveness of the proposed framework in enhancing the accuracy and reliability of LLM responses. The paper also analyzes the relationship between uncertainty and correctness, and shows that the logit output values of the LLM partially reflect inherent uncertainty.

**Strengths:**

- The paper proposes a remarkably simple method and is easy to understand. The technical details are clearly written and easy to follow.
- The paper presents evidence to two conclusions: 1) the logarithmic output values of the LLM partially reflect inherent uncertainty. 2) finetuning on calibration resulting in improved response accuracy.

**Weaknesses:**

- Evaluation lacks comparison with other uncertainty estimation methods [1,2]. It only compares different variants of its own framework.
- The claim "multi-inference-based uncertainty estimation methods need to get the answer set before calculating the uncertainty." is wrong. multi-inference-based uncertainty estimation methods (e.g. [3]) do not need correct answer. In fact, any uncertainty estimation method that requires the gold answer set is useless.
- It is not clear to me how uncertainty-aware correction works in SFT setting. It seems to me that the model never sees the uncertainty score during finetuning but is required to utilize the scores during inference. The discrepancy between finetuning and inference is not intuitive and not supported by empirical results.

[1] Can LLMs Express Their Uncertainty? An Empirical Evaluation of Confidence Elicitation in LLMs
[2] Just ask for calibration: Strategies for eliciting calibrated confidence scores from language models fine-tuned with human feedback.
[3] Semantic uncertainty: Linguistic invariances for uncertainty estimation in natural language generation.

**Questions:**

Please refer to the weakness section.

---

### Meta-Review · Area_Chair_basY · 2023-12-04

**Metareview:**

This paper introduces a method for using uncertainty within in-context learning, to allow LLMs to abstain from prediction if the model is too uncertain or "enhance" the prediction if it is confident.  Getting LLMs to express their uncertainty or use uncertainty to abstain from prediction is clearly timely, important and topical for ICLR.  Unfortunately, the reviewers unanimously agreed that the paper was not ready for publication at ICLR (with scores of 3, 3, 3).  The common concerns among the reviewers included missing discussion and citations of relevant literature, not thorough enough empirical evaluation (needs more baselines + benchmarks) and a lack of clarity.  On the positive side the reviewers found the method novel, interesting and simple (in a positive way) such that it could be easily deployed.  Therefore, it seems like this paper is a good start, but it would need substantial more work for a top conference publication.  Hopefully the reviews will be helpful for a future submission.

**Justification For Why Not Higher Score:**

Reviewers were unanimous about rejection.

**Justification For Why Not Lower Score:**

Reviewers were unanimous about rejection.

---

### Decision · Program_Chairs · 2024-01-16

Reject